# Development and Characterization of Two Wheat–Rye Introgression Lines with Resistance to Stripe Rust and Powdery Mildew

**DOI:** 10.3390/ijms252111677

**Published:** 2024-10-30

**Authors:** Yuzhou Ji, Guotang Yang, Xingfeng Li, Honggang Wang, Yinguang Bao

**Affiliations:** 1State Key Laboratory of Wheat Improvement, Shandong Agricultural University, Tai’an 271018, China; 2Agronomy College, Shandong Agricultural University, Tai’an 271018, China

**Keywords:** wheat, rye, wide hybridization, cytogenetic analysis, stripe rust, powdery mildew

## Abstract

Rye (*Secale cereale* L.) genes, which contribute to the tertiary gene pool of wheat, include multiple disease resistance genes useful for the genetic improvement of wheat. Introgression lines are the most valuable materials for wheat breeding because of their small alien segments and limited or lack of linkage drag. In the present study, wheat–rye derivative lines SN21627-2 and SN21627-6 were produced via distant hybridization. A genomic in situ hybridization analysis revealed that SN21627-2 and SN21627-6 lack alien segments, while a multi-color fluorescence in situ hybridization analysis detected structural changes in both introgression lines. At the seedling and adult plant stages, SN21627-2 and SN21627-6 were highly resistant to stripe rust and powdery mildew. Primers for 86 PCR-based landmark unique gene markers and 345 rye-specific SLAF markers were used to amplify SN21627-2 and SN21627-6 genomic DNA. Eight markers specific to rye chromosome 2R were detected in both introgression lines, implying these lines carry chromosome 2R segments with genes conferring stripe rust and powdery mildew resistance. Therefore, SN21627-2 and SN21627-6 are resistant to more than one major wheat disease, making them promising bridging parents for breeding disease-resistant wheat lines.

## 1. Introduction

Wheat (*Triticum aestivum* L.), which is one of the main cereal crops cultivated worldwide, provides more than 20% of the calories and protein consumed by humans [1]. Wheat stripe rust caused by *Puccinia striiformis* Westend. f. sp. *tritici* (*Pst*) and powdery mildew caused by *Blumeria graminis* (DC.) E.O. f. sp. *tritici* (*Bgt*) are devastating diseases threatening global wheat production [2,3]. Stripe rust and powdery mildew mainly damage wheat foliage and disrupt wheat photosynthesis activities, resulting in premature leaf senescence and impaired grain filling [4]. Stripe rust is prevalent in cool and humid regions, whereas powdery mildew has been detected mainly in relatively warm and dry areas in recent years [5]. Hence, breeding disease-resistant wheat cultivars is considered to be an economically viable and environmentally friendly method for controlling these two diseases. However, because of the rapid evolution of new pathogenic races, some widely used resistance genes can no longer protect plants from disease [6]. For example, *Pm8* from chromosome 1RS of Petkus rye is ineffective against new *Bgt* isolate No.9 collected in Sichuan, China [7]. Thus, there is an urgent need for research aimed at developing new disease-resistant germplasm and exploring novel resistance genes in wild wheat relatives to ensure these diseases can still be managed [8].

Rye (*Secale cereale* L., 2*n* = 2*x* = 14, RR) is a wild relative of common wheat that possesses many valuable genes for improving wheat disease resistance [9,10,11,12]. To date, 17 resistance genes have been identified and transferred into wheat: *Pm8*, *Pm17*, *Lr26*, *Sr31*, *Sr50*, *Sr36* and *Yr9* from chromosome 1R; *Pm7*, *Lr25*, *Lr45* and *Sr59* from 2R; *Sr27* from 3R; and *Pm20*, *Pm56, Yr83*, *PmTR1* and *PmTR3* from 6R [13,14,15]. Among these genes, *Pm8* and *Pm17* encode nucleotide-binding leucine-rich repeat (NLR) proteins [16,17]. *PmTR1* and *PmTR3* on rye chromosome 6R were isolated and identified as two NLR-encoding alleles [18]. *PmTR1* confers age-related resistance starting from the three-leaf stage, while *PmTR3* confers resistance at all growth stages. The stripe rust resistance gene *Yr83*, which is highly effective against Australian and Chinese *Pst* races, was physically mapped to fraction length 0.87–1.00 on the long arm of chromosome 6R on the basis of a fluorescence in situ hybridization (FISH) analysis and specific PCR marker amplification [19]. In addition to these previously reported genes, many resistance loci have been identified in different rye genome regions. For example, a powdery mildew resistance gene was physically mapped to the terminal region of chromosome 4RL using a new FISH map and 4R dissection lines [20].

With the development of cytogenetic methods for identifying alien segments, some wheat–rye addition, substitution and translocation lines have been obtained via chromosome engineering. For example, two complete sets of wheat–rye addition lines were established by wide crossing, chromosome doubling and backcrossing. These lines included a wheat–rye 3RL telosomic addition line highly resistant to Ug99 race PTKST [21]. A wheat–rye 4R disomic addition line (WR35), which was developed through distant hybridization, embryo rescue culture, chromosome doubling and backcrossing, may carry genes conferring resistance to wheat stripe rust, powdery mildew and sharp eyespot [22]. WR49-1, a wheat–rye 6R disomic addition line, is highly resistant to *Bgt* pathogens prevalent in China during the seedling and adult stages [23]. Using ^60^Coγ-ray irradiation, 164 wheat–rye translocation lines with chromosome 6R segments were developed, of which 106 exhibit all-stage resistance to powdery mildew [24]. Two new wheat–rye T1RS.1BL translocation lines, RT828-10 and RT828-11, were selected from the progeny of a cross between wheat cultivar Mianyang11-1 and rye variety Weining. Both lines are highly resistant to *Pst* and *Bgt* races that are unaffected by *Yr9* and *Pm8* [25].

In this study, two wheat–rye derivative lines, SN21627-2 and SN21627-6, were selected from the progeny derived from the hybridization between hexaploid triticale SN18730 (2*n* = 6*x* = 42, AABBRR) and common wheat cultivars Shannong 3050 (SN3050) and Gaoyou 5218 (GY5218). These lines were assessed in terms of their susceptibility to stripe rust and powdery mildew as well as their agronomic traits. Moreover, the chromosome compositions and variations in both lines were elucidated by conducting a cytogenetic analysis and amplifying molecular markers. The study data serve as important reference material for determining the utility of both wheat–rye derivative lines for breeding disease-resistant wheat varieties.

## 2. Results

### 2.1. Assessment of Stripe Rust Resistance

At the seedling stage, SN21627-2, SN21627-6 and their parents were inoculated with *Pst* race V26, and then their disease responses were assessed when the susceptible control Huixiangong (HXH) was thoroughly infected. The results showed that rye, SN18730, SN21627-2 and SN21627-6 were all immune, with an infection type (IT) score of 0, whereas the wheat parents SN3050 and GY5218 were susceptible (IT = 4) (Figure 1).

### 2.2. Assessment of Powdery Mildew Resistance

The resistance levels of SN21627-2, SN21627-6 and their parents to powdery mildew were recorded at the seedling stage (Figure 2, Table 1). Among the samples inoculated with *Bgt* race E09, the susceptible control HXH and wheat cultivars GY5218 and SN3050 had a higher IT score (IT = 4) than rye, SN18730 and SN21627-2 (IT = 0; i.e., immune) and SN21627-6 (IT = 1; i.e., highly resistant). At the adult stage, SN21627-2, SN21627-6, rye and SN18730 were all immune to powdery mildew (IT = 0), while wheat varieties HXH, GY5218 and SN3050 were highly susceptible (IT = 4) (Table 1).

### 2.3. Analysis of Chromosome Compositions

A genomic in situ hybridization (GISH) analysis revealed that SN21627-2 and SN21627-6 carry 42 wheat chromosomes, with no detectable signal for alien segments (Figure 3A,C), while SN18730 carries 28 wheat chromosomes and 14 rye chromosomes (Figure 3E). After removing GISH signals, the same slides were subjected to a multi-color fluorescence in situ hybridization (mc-FISH) analysis, after which the mc-FISH patterns of SN21627-2 and SN21627-6 were compared with those of their parents GY5218 (Figure 3B,D) and SN18730 (Figure 3F). For chromosomes 1A, 2A, 4A, 6A, 7A, 1B, 2B, 3B, 4B, 5B, 6B, 7B and 2D, different signals were detected in the two introgression lines (Figure 4). This suggests that chromosome structures changed during the formation of these two introgression lines.

### 2.4. Molecular Marker Analysis

To analyze the rye genetic material in SN21627-2 and SN21627-6, 86 PCR-based landmark unique gene (PLUG) markers and 345 rye-specific specific-locus amplified fragment (SLAF) markers were amplified using genomic DNA (gDNA) extracted from SN21627-2, SN21627-6 and their parents (Figure 5). According to the results, the primers for two PLUG markers and seven SLAF markers specific to rye chromosome 2R amplified the same fragments for rye, SN18730, SN21627-2 and SN21627-6 but not for the common wheat parents GY5218 and SN3050 (Appendix A), implying that SN21627-2 and SN21627-6 contain gDNA from rye chromosome 2R.

## 3. Discussion

Wild relatives of wheat are valuable resources because of their resistance to biotic and abiotic stresses. To date, useful genes from many wild relatives, including rye, *Thinopyrum ponticum*, *Thinopyrum intermedium*, *Dasypyrum villosum* and *Agropyrum cristatum*, have been transferring their useful genes into common wheat. Rye, a naturally cross-pollinating relative of wheat, has been exploited to improve wheat varieties since the last century. Several essential resources, such as wheat–rye addition, substitution and translocation lines, have been developed via distant hybridization and chromosome engineering. In these lines, the 1RS.1BL translocation represents the most successful use of rye chromosomes by wheat breeders. Notably, approximately 30% of wheat cultivars released after 2000 carry the 1RS.1BL translocation [26]. The successful application of the 1RS.1BL translocation may be attributed to the following two reasons: chromosome arm 1RS contains many disease resistance genes, and this translocation is not associated with any linkage drag adversely affecting grain yield.

Rye chromosome 2R reportedly carries disease resistance genes. WR02-145 lines, which were identified as wheat–rye 2R (2D) disomic chromosome substitution lines, are resistant to powdery mildew isolates prevalent in northern China [27]. Zhuang et al. (2011) mapped a powdery mildew resistance gene on chromosome 2RL of Chinese rye cultivar Jingzhouheimai according to a linkage analysis [28]. In the current study, SN21627-2 and SN21627-6 were resistant to *Pst* race V26 at the seedling stage and highly resistant to powdery mildew at all growth stages. In terms of pedigree, the resistance genes in both introgression lines may be encoded in their alien DNA sequences. On the basis of rye-specific molecular marker amplification, SN21627-2 and SN21627-6 contain genetic components from rye chromosome 2R. Therefore, we speculate that rye chromosome 2R includes stripe rust and powdery mildew resistance genes.

Wheat–rye addition, substitution and translocation lines usually contain undesirable chromatin that can affect wheat yield and quality. Therefore, wheat–rye introgression lines are considered to be ideal germplasm resources because of their relatively short alien segments and limited linkage drag. For example, wheat cultivar Xiaoyan 6, which was identified from among the hybrids generated by a cross between wheat and *Th. ponticum*, produces high-quality grains while also being resistant to multiple stresses and exhibiting good agronomic traits. More than 80 cultivars have been developed from Xiaoyan 6. The accumulated cultivation area for these cultivars exceeds 300 million mu in China [29]. Wheat–rye introgression lines SN21627-2 and SN21627-6, which were developed in the present study, are resistant to stripe rust and powdery mildew. Additionally, compared to recurrent parent GY5218, SN21627-2 is shorter, while SN21627-6 has a longer main panicle and produces more kernels per main panicle (Figure 6). The yield potential and grain quality of SN21627-2 and SN21627-6 are currently being determined in field trials. Thus, wheat–rye introgression lines SN21627-2 and SN21627-6 are resistant to stripe rust and powdery mildew, but they also have relatively short alien segments and exhibit good agronomic performance, making them potentially useful for breeding new disease-resistant wheat varieties.

In summary, we developed two wheat–rye introgression lines (SN21627-2 and SN21627-6) resistant to stripe rust and powdery mildew. The related resistance genes may have been derived from rye chromosome 2R. Because of their good disease resistance and agronomic performance, these two introgression lines may be useful for breeding novel wheat varieties resistant to stripe rust and powdery mildew.

## 4. Materials and Methods

### 4.1. Plant Materials

Wheat–rye BC_1_F_5_ derivative lines (SN21627-2 and SN21627-6) were developed from the SN18730/SN3050//GY5218 cross. Specifically, the common wheat variety SN3050 was hand-pollinated with pollen from SN18730 to produce F_1_ interspecific hybrids, which were then pollinated using pollen from the high-quality wheat cultivar GY5218. Selected lines with good agronomic traits were then backcrossed three times with the recurrent parent GY5218. Finally, stable lines were selected according to their yield performance. Hexaploid triticale SN18730, common wheat GY5218, SN3050, Yannong 15 (YN15) and HXH (wheat cultivar used as a susceptible control) were preserved in our laboratory.

### 4.2. Stripe Rust Resistance Evaluation

When the first leaves were fully expanded, inoculation was performed by spraying the mixture of urediniospores of *Pst* race CYR34 and talcum powder at a ratio of 1:2 onto the seedling leaves of SN21627-2, SN21627-6, their parents and HXH (10 plants each) at the State Key Laboratory of Wheat Improving, Shandong Agricultural University. Inoculated seedlings were incubated in a darkened chamber at 20 °C with 100% relative humidity (RH) for 24 h. They were then transferred to a greenhouse and maintained at 14–16 °C with a 14 h light/10 h dark cycle and 70% RH. When HXH plants were heavily infected (i.e., 2 weeks post-inoculation), IT scores were recorded (0–4 scale). In detail, “0” represents immunity, no visible uredinia and necrosis on leaves; “0;” represents nearly immune, no visible uredinia with hypersensitive flecks on leaves; “1” represents highly resistant, small uredinia surrounded by necrosis; “2” represents moderate resistance, small-to-medium uredinia surrounded by necrosis; “3” represents moderate susceptible, medium uredinia without chlorosis or necrosis; and “4” represents highly susceptible, large uredinia without chlorosis or necrosis. Plants with IT scores of 0–2 and 3–4 were considered resistant and susceptible, respectively [30].

### 4.3. Powdery Mildew Resistance Evaluation

*Bgt* race E09 was used to evaluate powdery mildew resistance because it is a representative highly virulent isolate. Using the sweeping method, HXH, SN21627-2, SN21627-6 and their parents (10 plants each) were inoculated with E09 at the two-leaf stage. IT scores were recorded at 10–12 days post-inoculation as described by Si et al. (1992) [31], where 0 = no visible symptoms, 0; = necrosis flecks, 1 = necrosis with low sporulation, 2 = necrosis with moderate sporulation, 3 = no necrosis with high sporulation and 4 = no necrosis with full sporulation. Plants with IT scores of 0–2 represent resistant, whereas those with 3–4 represent susceptible. In addition, field-grown plants were naturally infected at the booting and grain-filling stages, and then powdery mildew resistance was examined during the adult stage. Plants were grown at the Experimental Station of Shandong Agricultural University. IT scores were recorded (0–4 scale), where 0 = no visible symptoms in whole plant, 0; = necrosis flecks in whole plant, 1 = sporulation appears on basal leaves, 2 = sporulation appears on basal to middle leaves, 3 = sporulation on middle to upper leaves and 4 = sporulation appears in spike region. Plants with IT scores of 0–2 and 3–4 were considered resistant and susceptible, respectively.

### 4.4. Chromosome Preparation

Chromosomes were prepared as described by Kato et al. (2004), with minor modifications [32]. Briefly, five seeds per line were germinated on moistened filter paper in a 23 °C incubator for 2 days. Root tips that were 1–3 cm long were cut, pretreated with N_2_O under 10 atm pressure for 2 h, fixed in 90% acetic acid for 8 min and then digested with 2% cellulase and 1% pectinase. The root tip meristem was rinsed with 75% ethanol, mashed and diluted with 100% glacial acetic acid. A 10 µL aliquot of this mixture was dropped onto the center of a slide. Slides with good mitotic phases were used for the subsequent cytogenetic analysis.

### 4.5. Cytogenetic Identification

Sequential GISH and mc-FISH analyses were performed as described by Du et al. (2017) [33,34]. For the GISH analysis, rye gDNA labeled with Alexa Fluor-488-dUTP (green) was used as a probe, whereas YN15 gDNA was used for blocking (50:1 block–probe ratio). The following eight probes were synthesized by Shanghai Sangon Biotechnology Co., Ltd. (Shanghai, China), and used for the mc-FISH analysis: AFA-3 (red), AFA-4 (red), pAs1-1 (red), pAs1-3 (red), pAs1-4 (red), pAs16 (red), pSc119.2-1 (green) and (GAA)10 (green). After the hybridization, slides were washed in a 2× Saline Sodium Citrate and then counterstained with 4, 6-diamidino-2-phenylindole. Good hybridization signals were examined using an Olympus BX60 fluorescence microscope, photographed using a charge-coupled device camera and analyzed using CellSens Standard 1.12 (Olympus, Tokyo, Japan). Wheat chromosomes were identified according to CS standard spectral bands [35].

### 4.6. Molecular Marker Analysis

A modified cetyltrimethylammonium bromide method was used to extract gDNAs from young leaf tissue collected from SN21627-2, SN21627-6 and their parents [36]. To examine whether SN21627-2 and SN21627-6 contain genetic material from the rye genome, primers for 86 PLUG markers and 345 SLAF markers specific for rye were used to amplify the extracted gDNAs [37,38]. The 10 µL reaction mixture included 40 ng gDNA, 2 µM forward and reverse primers, 2.5 mM dNTPs, 2.5 mM MgCl_2_, 1× PCR buffer (10 mM Tris-HCl, pH 8.5, 50 mM KCl) and 0.5 U Taq DNA polymerase. The PTC-200 thermal cycler (Bio-Rad, Hercules, CA, USA) and the following program were used for the PCR amplification: 94 °C for 5 min; 36 cycles of 94 °C for 30 s, 55–60 °C (according to the primer annealing temperature) for 30 s and 72 °C for 30 s; and 72 °C for 10 min before cooling to 4 °C. PCR products were separated by 2% agarose gel electrophoresis and then photographed using the Tanon 1600 Gel Image System (Tanon, Shanghai, China).

## Figures and Tables

**Figure 1 ijms-25-11677-f001:**
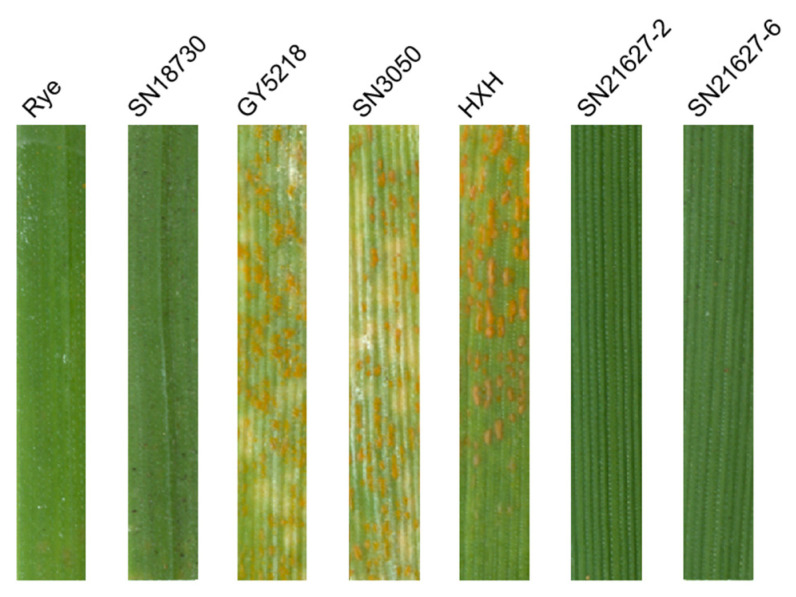
The seedling response of SN21627-2, SN21627-6 and their parents to *Pst* race V26.

**Figure 2 ijms-25-11677-f002:**
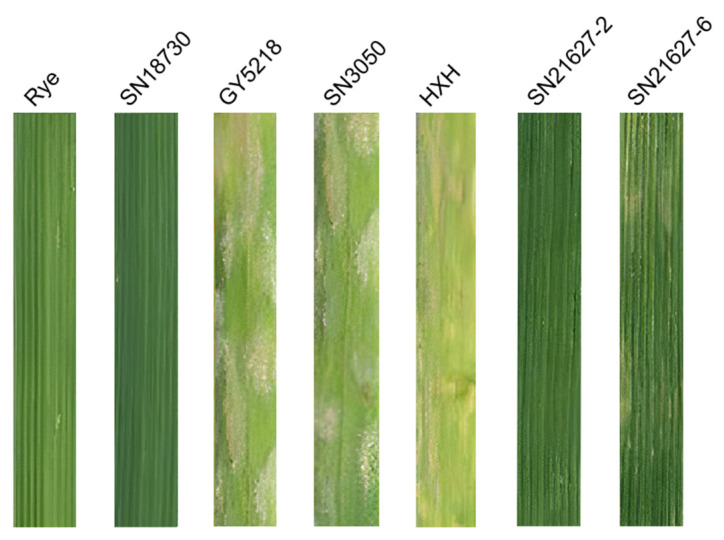
The seedling response of SN21627-2, SN21627-6 and their parents to *Bgt* race E09.

**Figure 3 ijms-25-11677-f003:**
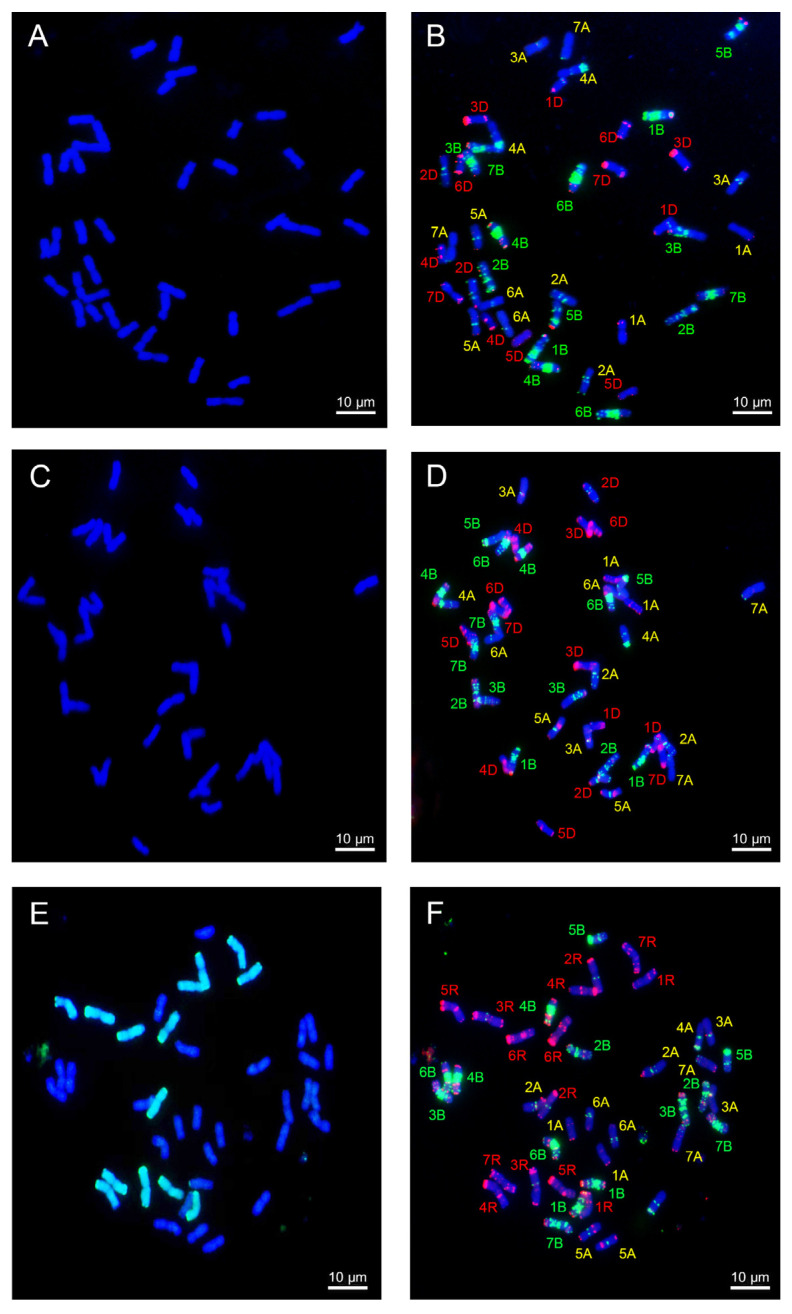
Genomic in situ hybridization (GISH) and fluorescence in situ hybridization (FISH) analyses of chromosomes of SN21627-2 (**A**,**B**), SN21627-6 (**C**,**D**) and SN18730 (**E**,**F**).

**Figure 4 ijms-25-11677-f004:**
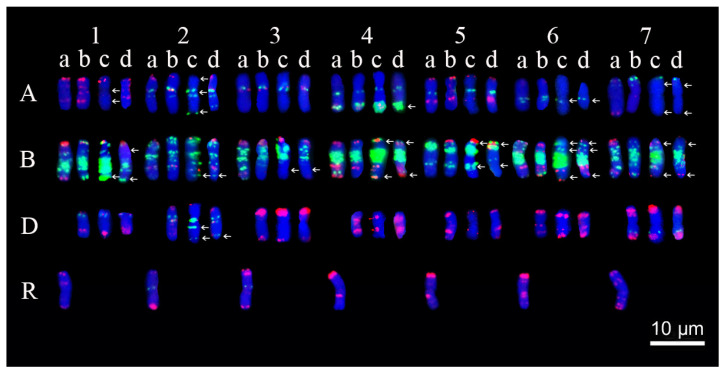
Chromosome structural variations in SN21627-2 and SN21627-6. a: SN18730; b: GY5218; c: SN21627-2; d: SN21627-6. Arrows indicate the chromosome structural variations. The numbers “1–7” indicate wheat homoeologous groups 1–7 and the letters “A, B, D, R” indicate wheat and rye genomes.

**Figure 5 ijms-25-11677-f005:**
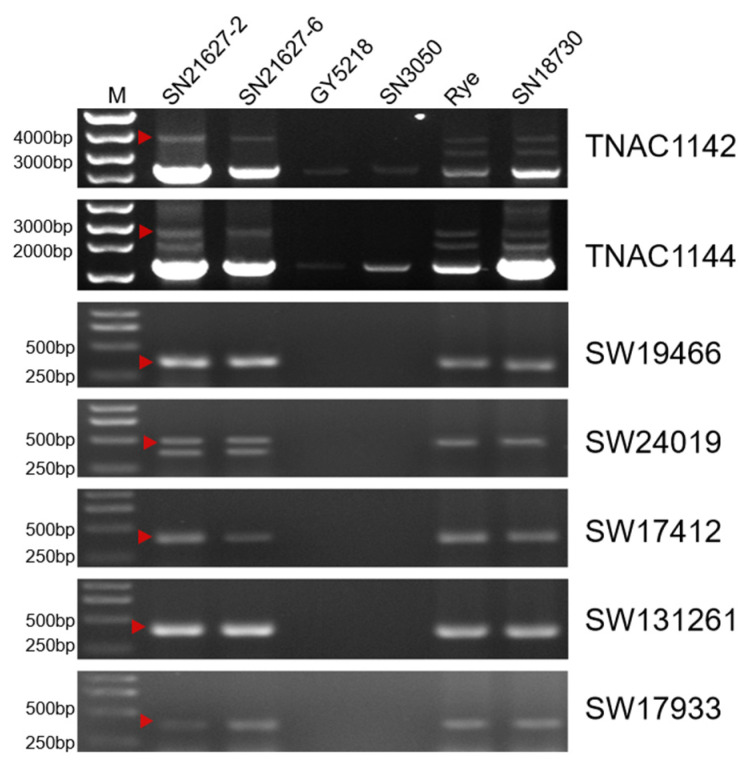
Molecular marker analysis of SN21627-2, SN21627-6 and their parents. Triangles indicate specific bands of rye.

**Figure 6 ijms-25-11677-f006:**
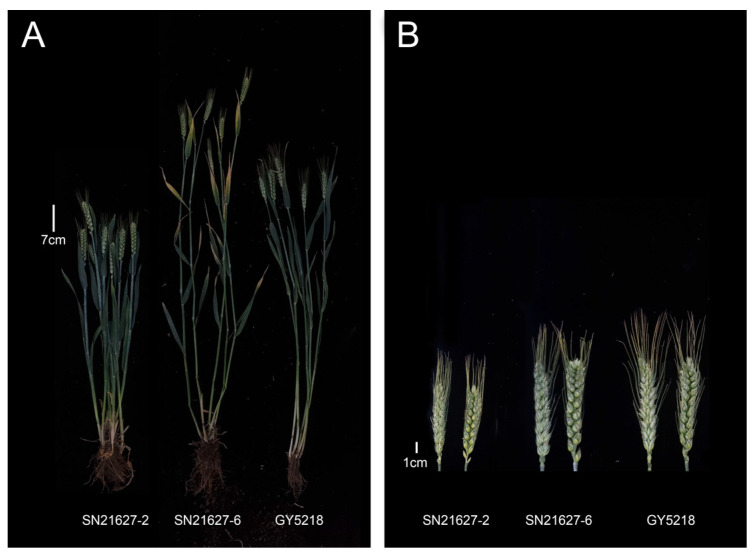
Plant (**A**) and spike (**B**) traits of SN21627-2 and SN21627-6.

**Table 1 ijms-25-11677-t001:** The seedling and adult disease response of wheat–rye derivative lines SN21627-2 and SN21627-6 and their parents to *Bgt* isolate E09.

Material	Powdery Mildew	
	Seedling Stage	Adult Stage
RYE	0	0
SN18730	0	0
SN21627-2	0	0
SN21627-6	1	0
GY5218	4	4
SN3050	4	4
HXH	4	4

## Data Availability

All data supporting the findings of this study are available within this article.

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
