# Peer review of "Development and Characterization of Two Wheat–Rye Introgression Lines with Resistance to Stripe Rust and Powdery Mildew"

_ijms, 2024, doi:10.3390/ijms252111677_

Round 1
Reviewer 1 Report
Comments and Suggestions for Authors
The manuscript by Ji and colleagues describes the development of two wheat-rye derivative lines that display resistance to stripe rust and powdery mildew. While these introgression lines are beneficial for wheat breeding in developing resistant cultivars against these diseases, I have several concerns outlined below:
1. The authors should include more details about the resistance evaluation experiment to ensure that it can be replicated by other researchers. For example, how was the resistance evaluation experiment performed? Were spores of the corresponding strain used for inoculation on plant leaves? What was the concentration of the spores? Additionally, the criteria used for determining plant infection types need to be clarified.
2. What is the key takeaway from Figure 3?
3. Many terms and analyses mentioned in the study (e.g., mc-FISH analysis, GISH signals, alien signal) are not explained when they first appear, making the manuscript difficult to understand.
4. Figure 4: The arrows are too small to be seen clearly. What do the different colors represent? Apart from chromosomes 1A and 5D of SN21627-2 and chromosome 4A of SN21627-6, it appears there are additional chromosomal differences between the two lines and GY5218. Why were these differences not mentioned in the main text? Furthermore, why do the authors only compare the two lines with their recurrent parent, GY5218?
5. What does "PLUG marker" refer to? Are the gel images in Figure 5 showing PCR results from PLUG markers or PCR-based markers? Why are there multiple bands in some of the gel images? Are these a result of non-specific binding? Since the authors tested 86 PLUG primer pairs and 345 rye-specific PCR-based markers, they should include the remaining gel images in the supporting information.
6. In the statement, “The results of rye-specific molecular marker amplification indicated that SN21627-2 and SN21627-6 contained genetic components from rye chromosome 2RL. Therefore, we can speculate that rye chromosome 2RL carries the stripe rust and powdery mildew resistance gene,” the difference between rye chromosome 2R and 2RL needs to be clarified. Additionally, the authors should perform further experiments to support this speculation.
Comments on the Quality of English LanguageThe overall quality of the English language is good and only requires minor revisions
Author Response
Dear editor:
Thank you very much for sending us the reviewers’ comments on our manuscript (ijms-3192175) entitled “Development and characterization of two wheat–rye introgression lines with resistance to stripe rust and powdery mildew”. Those comments are all valuable and helpful for revising and improving our paper. The following is the answers and revisions we have made in response to the reviewers’ questions and suggestions on an item by item basis. We have checked the paper again for English editing and the changes to the manuscript were also highlighted in yellow text.
Thanks again for your consideration of our manuscript. We are willing to respond to any further questions and comments that you may have.
Yours sincerely
Guotang Yang & Yinguang Bao
In response to the reviewers’ comments: (reviewers’ comments are marked in bold)
- The authors should include more details about the resistance evaluation experiment to ensure that it can be replicated by other researchers. For example, how was the resistance evaluation experiment performed?
Answer: Thanks for your suggestions. Related details about the resistance evaluation experiment have been added.
4.2 Stripe rust resistance evaluation
When HXH plants were heavily infected (i.e., 2 weeks post-inoculation), IT scores were recorded (0–4 scale). In detail, “0” represents immunity, no visible uredinia and necrosis on leaves; “0;” represents nearly immune, no visible uredinia with hypersensitive flecks on leaves; “1” represents highly resistant, small uredinia surrounded by necrosis; “2” represents moderate resistance, small-to-medium uredinia surrounded by necrosis; “3” represents moderate susceptible, medium uredinia without chlorosis or necrosis; “4” represents highly susceptible, large uredinia without chlorosis or necrosis. Plants with IT scores of 0–2 and 3–4 were considered resistant and susceptible, respectively (lines 177-185).
4.3 Powdery mildew resistance evaluation
Using the sweeping method, HXH, SN21627-2, SN21627-6, and their parents (10 plants each) were inoculated with E09 at the two-leaf stage. IT scores were recorded at 10–12 days post-inoculation as described by Si et al. (1992) [31], where 0 = no visible symptoms, 0; = necrosis flecks, 1 = necrosis with low sporulation, 2 = necrosis with moderate sporulation, 3 = no necrosis with high sporulation, 4 = no necrosis with full sporulation. Plants with IT scores of 0–2 represent resistant, whereas 3–4 represent susceptible (lines 188-194).
IT scores were recorded (0-4 scale), where 0 = no visible symptoms in whole plant, 0; = necrosis flecks in whole plant, 1 = sporulation appears on basal leaves, 2 = sporulation appears on basal to middle leaves, 3 = sporulation appears middle to upper leaves, 4 = sporulation appears spike region. Plant with IT scores of 0–2 and 3–4 were considered resistant and susceptible, respectively (lines 196-201).
- Were spores of the corresponding strain used for inoculation on plant leaves?
Answer: Yes. Related description has been added. When first leaves were fully expanded, inoculation was performed by spraying the mixture of urediniospores of Pst race CYR34 and talcum powder at a ratio of 1:2 onto the seedling leaves of SN21627-2, SN21627-6, their parents, and HXH (10 plants each) at the State Key Laboratory of Wheat Improving, Shandong Agricultural University (lines 171-174).
- What was the concentration of the spores?
Answer: The inoculation was performed by spraying the mixture of urediniospores of Pst race CYR34 and talcum powder at a ratio of 1:2 onto the seedling leaves (lines 171-173). The inoculation of Bgt race was used the sweeping method. Thus, the concentration of the spores was unknown.
- The criteria used for determining plant infection types need to be clarified.
Answer: The criteria of stripe rust and powdery mildew evaluation have been added. When HXH plants were heavily infected (i.e., 2 weeks post-inoculation), IT scores were recorded (0–4 scale). In detail, “0” represents immunity, no visible uredinia and necrosis on leaves; “0;” represents nearly immune, no visible uredinia with hypersensitive flecks on leaves; “1” represents highly resistant, small uredinia surrounded by necrosis; “2” represents moderate resistance, small-to-medium uredinia surrounded by necrosis; “3” represents moderate susceptible, medium uredinia without chlorosis or necrosis; “4” represents highly susceptible, large uredinia without chlorosis or necrosis. Plants with IT scores of 0–2 and 3–4 were considered resistant and susceptible, respectively (lines 177-185).
IT scores were recorded at 10–12 days post-inoculation as described by Si et al. (1992) [31], where 0 = no visible symptoms, 0; = necrosis flecks, 1 = necrosis with low sporulation, 2 = necrosis with moderate sporulation, 3 = no necrosis with high sporulation, 4 = no necrosis with full sporulation. Plants with IT scores of 0–2 represent resistant, whereas 3–4 represent susceptible. In addition, field-grown plants were naturally infected at the booting and grain-filling stages and then powdery mildew resistance was examined during the adult stage. Plants were grown at the Experimental Station of Shandong Agricultural University. IT scores were recorded (0-4 scale), where 0 = no visible symptoms in whole plant, 0; = necrosis flecks in whole plant, 1 = sporulation appears on basal leaves, 2 = sporulation appears on basal to middle leaves, 3 = sporulation appears middle to upper leaves, 4 = sporulation appears spike region. Plant with IT scores of 0–2 and 3–4 were considered resistant and susceptible, respectively (lines 190-201).
- What is the key takeaway from Figure 3?
Answer: The result of GISH analysis (A, C) showed that no alien fluorescence signals detected in chromosomes of SN21627-2 and SN21627-6 with rye gDNA labeled with Alexa Fluor-488-dUTP (green) as a probe and YN15 gDNA as a blocking, indicating that both lines were wheat–rye introgression lines. The mc-FISH analysis was used to compare the different signals between introgression line and its recurrent parent and seek the chromosome structure changed during the introgression line formation.
- Many terms and analyses mentioned in the study (e.g., mc-FISH analysis, GISH signals, alien signal) are not explained when they first appear, making the manuscript difficult to understand.
Answer: Thanks for your suggestions. We have added the full name when they first appear. In this study, two wheat–rye derivative lines, SN21627-2 and SN21627-6, were selected from the progeny derived from the hybridization between hexaploid triticale SN18730 (2n = 6x = 42, AABBRR) and common wheat cultivars Shannong 3050 (SN3050) and Gaoyou 5218 (GY5218) (lines 73-76). A genomic in situ hybridization (GISH) analysis revealed that SN21627-2 and SN21627-6 carry 42 wheat chromosomes, with no detectable signal for alien segments (Fig. 3A, 3C), while SN18730 carries 28 wheat chromosomes and 14 rye chromosomes (Fig. 3E). After removing GISH signals, the same slides were subjected to a multi-color fluorescence in situ hybridization (mc-FISH) analysis, after which the mc-FISH patterns of SN21627-2 and SN21627-6 were compared with those of their parents GY5218 (Fig. 3B, 3D) and SN18730 (Fig. 3F) (lines 97-103). To analyze the rye genetic material in SN21627-2 and SN21627-6, 86 PCR-based landmark unique gene (PLUG) markers and 345 rye specific specific-locus amplified fragment (SLAF) markers were amplified using genomic DNA (gDNA) extracted from SN21627-2, SN21627-6 and their parents (Fig. 5) (lines 107-110).
- Figure 4: The arrows are too small to be seen clearly. What do the different colors represent?
Answer: We have adjusted the arrow size in Figure 4. Eight probes were used for the mc-FISH analysis. Red colors indicate AFA and pAs1 tandem repeat sequence signals, green colors indicate pSc119.2 and (GAA)10 tandem repeat sequence signals.
- Apart from chromosomes 1A and 5D of SN21627-2 and chromosome 4A of SN21627-6, it appears there are additional chromosomal differences between the two lines and GY5218. Why were these differences not mentioned in the main text?
Answer: The reason for these FISH pattern difference is that chromosome structures changed during the formation of these two introgression lines (lines 104-105).
- Why do the authors only compare the two lines with their recurrent parent, GY5218?
Answer: Because the GY5218 is the recurrent parent of SN21627-2 and SN21627-6, we mainly compare the chromosome karyotype between two lines and their recurrent parent. In order to find the different between two lines and hexaploidy triticale SN18730, the chromosome karyotype of SN18730 was added in Fig. 3 and Fig. 4.
- What does "PLUG marker" refer to?
Answer: PLUG marker refers to PCR-based landmark unique gene marker. We have added the full name in the paper. To analyze the rye genetic material in SN21627-2 and SN21627-6, 86 PCR-based landmark unique gene (PLUG) markers and 345 rye specific PCR-based markers were amplified using genomic DNA (gDNA) extracted from SN21627-2, SN21627-6 and their parents (lines 107-110).
- Are the gel images in Figure 5 showing PCR results from PLUG markers or PCR-based markers?
Answer: In Fig. 5, two PCR-based landmark unique gene (PLUG) markers (TNAC1142 and TNAC1144) and five specific-locus amplified fragment (SLAF) markers (SW17933, SW19466, SW24019, SW131261, SW17412) were amplified specific DNA bands in rye genome.
- Why are there multiple bands in some of the gel images? Are these a result of non-specific binding?
Answer: Because of nonspecific amplification, multiple bands were appeared in some of gel images. However, main bands are existed in every gel images. Thus, there are a result of specific binding.
- Since the authors tested 86 PLUG primer pairs and 345 rye-specific PCR-based markers, they should include the remaining gel images in the supporting information.
Answer: A total of 86 PLUG and 345 SLAF markers were used in amplification, only two PLUG and seven SLAF markers could be amplified in rye genome. Specific bands were not detected in both introgression lines in the remaining gel images and the picture number is so many. Thus, these gel images were not been provided.
- In the statement, “The results of rye-specific molecular marker amplification indicated that SN21627-2 and SN21627-6 contained genetic components from rye chromosome 2RL. Therefore, we can speculate that rye chromosome 2RL carries the stripe rust and powdery mildew resistance gene,” the difference between rye chromosome 2R and 2RL needs to be clarified. Additionally, the authors should perform further experiments to support this speculation.
Answer: We are very sorry about the error. We have changed rye chromosome 2RL to 2R (lines 137). Because the screened specific markers belong to chromosome 2R and the wheat parents are susceptible to stripe rust and powdery mildew. Thus, we can speculate that rye chromosome 2R carries the related resistance gene. Further experiments, including genetic population construction and fine mapping, will performed to support this speculation.

Reviewer 2 Report
Comments and Suggestions for Authors
This article reports on two wheat-rye introgression lines, SN21627-2 and SN21627-6, which exhibit resistance to stripe rust and powdery mildew. Specific molecular markers were used to verify that the resistance genes may originate from the introgression of rye chromosome 2R. Due to their excellent disease resistance and agronomic traits, these lines hold potential as valuable breeding materials for wheat disease resistance, making this study highly significant for wheat breeding research. However, there are still some areas that require improvement:
1. In Figure 3, the corresponding positions of the chromosomes in panels A and B, as well as C and D, should be as consistent as possible, and the size of the chromosomes and scales should also be uniform. It is recommended to provide in situ hybridization results for the parental materials.
2. In Figure 4, the position of the introgression signals indicated by white arrows, such as on 5D, needs to be verified.
3. The description of Figure 5 should specify the chromosome and position information for the molecular markers used in the screening, and it is suggested to include detailed information about these specific molecular markers.
4. In Results Section 2.4, the presence of rye 2R sequences is mentioned, but their exact distribution across the A, B, and D genomes should be clarified.
The conclusions section needs further elaboration, and some figures and result descriptions need to be refined to improve the accuracy and clarity of the article.
Author Response
Dear editor:
Thank you very much for sending us the reviewers’ comments on our manuscript (ijms-3192175) entitled “Development and characterization of two wheat–rye introgression lines with resistance to stripe rust and powdery mildew”. Those comments are all valuable and helpful for revising and improving our paper. The following is the answers and revisions we have made in response to the reviewers’ questions and suggestions on an item by item basis. We have checked the paper again for English editing and the changes to the manuscript were also highlighted in yellow text.
Thanks again for your consideration of our manuscript. We are willing to respond to any further questions and comments that you may have.
Yours sincerely
Guotang Yang & Yinguang Bao
In response to the reviewers’ comments: (reviewers’ comments are marked in bold)
- In Figure 3, the corresponding positions of the chromosomes in panels A and B, as well as C and D, should be as consistent as possible, and the size of the chromosomes and scales should also be uniform. It is recommended to provide in situ hybridization results for the parental materials.
Answer: Thanks for your suggestion. We have adjusted and added the parental material (hexaploidy triticale SN18730) in Fig. 3.
- In Figure 4, the position of the introgression signals indicated by white arrows, such as on 5D, needs to be verified.
Answer: In Fig. 4, the arrows indicate different signals between two introgression lines and their parents rather than introgression signals, suggesting that chromosome structure was changed during the formation of SN21627-2 and SN21627-6.
- The description of Figure 5 should specify the chromosome and position information for the molecular markers used in the screening, and it is suggested to include detailed information about these specific molecular markers.
Answer: The sequence information of these seven markers have been added in supplement table. We only used these specific markers to amplify the gDNA of both introgression lines and the amplification results indicated that two lines may carries rye chromosome 2R sequences. The amplified products of these molecular markers will be sequenced and compared the rye genome to determine the position information.
- In Results Section 2.4, the presence of rye 2R sequences is mentioned, but their exact distribution across the A, B, and D genomes should be clarified.
Answer: Because the lines SN21627-2 and SN21627-6 were regarded as introgressions, no alien signals were detected using GISH analysis. Thus, the molecular marker analysis was used to detect the alien DNA sequence. The result of molecular marker amplification indicates that rye chromosome 2R sequences are existed in both introgression lines. However, the locations of these rye sequences are difficult to determine.
- The conclusions section needs further elaboration, and some figures and result descriptions need to be refined to improve the accuracy and clarity of the article.
Answer: Thanks for your opinions. The related figures, methods, results and conclusions have been modified in this latest version of the manuscript. For example, Fig. 3 and Fig. 4 have been replaced the new ones. Disease evaluation details and result descriptions have been added in M&M and results sections.
